# Thinning Promotes Soil Phosphorus Bioavailability in Short-Rotation and High-Density *Eucalyptus grandis* × *E. urophylla* Coppice Plantation in Guangxi, Southern China

Xiangsheng Xiao [1,2], Izhar Ali [1,2], Xu Du [1,2], Yuanyuan Xu [1,2], Shaoming Ye [1,2] and Mei Yang [1,2,*]

1   Guangxi Colleges and Universities Key Laboratory for Cultivation and Utilization of Subtropical Forest Plantation, Guangxi University, Nanning 530004, China; 2109302017@st.gxu.edu.cn (X.X.); izharali@gmail.com (I.A.); 1809303003@st.gxu.edu.cn (X.D.); yuanyuanxu_2016@163.com (Y.X.); yshaoming@163.com (S.Y.)
2   Key Laboratory of National Forestry and Grassland Administration on Cultivation of Fast-Growing Timber in Central South China, College of Forestry, Guangxi University, Nanning 530004, China
*   Correspondence: fjyangmei@126.com

**Abstract:** Thinning can improve soil nutrient supply, but the effects of thinning on soil phosphorus (P) contents and bioavailable mechanisms in high-density and short-rotation *Eucalyptus* coppice forests are not well reported. Therefore, we conducted five intensities of thinning treatments, which were 83% (283 tree ha$^{-1}$, T1), 66% (566 tree ha$^{-1}$, T2), 50% (833 tree ha$^{-1}$, T3), 33% (1116 tree ha$^{-1}$, T4), and 0% (1665 tree ha$^{-1}$) in a 2nd 6-year-old *E. grandis* × *E. urophylla* coppice plantation with 8 years as a rotation, investigated soil nutrient contents, microbial biomass, and extracellular enzyme activities of 0–20 and 20–40 cm soil layers after two years of thinning, and analyzed the relationship between available phosphorus (AP) and other indicators. The results showed that soil total phosphorus (TP) contents in 2nd *Eucalyptus* coppice plantations were lower than in native forest ecosystems, but T1 significantly increased ($p < 0.05$) TP by 81.42% compared to CK of 0–20 cm, whereas T2 and T3 improved available phosphorus (AP) by 86.87%–212.86% compared to CK. However, soil organic carbon (SOC), dissolved organic carbon (DOC), total nitrogen (TN), and alkaline hydrolysable nitrogen (AN) were not significantly different ($p < 0.05$) among all treatments. According to the analysis, soil TP contents were significantly positively related ($p < 0.001$) to SOC; soil total nutrients and DOC contents had the highest standardized total effect on AP; meanwhile, the quotient of microbial biomass directly conducted soil AP contents. These results highlighted that thinning can be used to alleviate soil P shortages by promoting multinutrient and biological cycles in *Eucalyptus* coppice forests.

**Keywords:** *Eucalyptus* plantation; thinning; soil nutrients; phosphorus bioavailability; microbial biomass; extracellular enzyme activities



## 1. Introduction

In China, the Guangxi area is a major *Eucalyptus* plantation area, with approximately 4.5 million hectares accounting for 45% of the total *Eucalyptus* plantation area in this country [1]. To obtain the maximum profit, the high-density (over 1110 tree ha$^{-1}$) and short-rotation (5–10 years) management modes have become increasingly common in *Eucalyptus* plantation management in Guangxi. This plantation ecosystem suffers severe impacts from human activities and then leads to weaker ecological and social functions than native forest ecosystems [2], such as a decrease in the diversity of understory plant communities, degradation of aggregates, nutrient losses, etc. [3–6]. Meanwhile, some improved *Eucalyptus* species, which are usually used in plantations, require 2–4 times more nutrients than what is returned by litter in this mode [7–9]. Therefore, the contradiction between declining soil fertility and the rapid growth of *Eucalyptus* is fiercer after the

intraspecific competition starts in this silvicultural system, suggesting some stand density control technologies are necessary in high-density *Eucalyptus* plantations.

Thinning decreases stand and canopy densities and can (1) relieve nutrients and space competition in high-density plantations, (2) promote plant regeneration, resumption, and litter input [10,11], and (3) ultimately benefit soil nutrient accumulation, cycling, and bioavailability [12–15]. In a previous study, the functions of thinning in sustainable management of *Eucalyptus* plantations were discussed, which included improved fertilizer utilization efficiency, increased stand size classes, and enhanced productivity after thinning to mitigate land degradation [16]. However, most thinning research has focused on coniferous timber plantations, such as Chinese fir (*Cunninghamia lanceolata*) and Masson pine (*Pinus massoniana*) in southern China [17,18]. As a result, it remains unclear how thinning affects the soil nutrient characteristics and biochemical cycles in high-density and short-rotation *Eucalyptus* plantations.

In low-latitude regions, soil bioavailable P is relatively scarce due to leach and inorganic P adsorbed by Fe and Al ions, which makes P often the limiting element for the structure and function of forest ecosystems in tropical and subtropical areas [19–21]. Although P fertilizer input is a common method for supplying soil P in short-rotation plantations, the bioavailable P fraction fixed in acid soil is the main reason for low plant availability after fertilizing [22,23]. This implies that soil TP may be sufficient, but very low bioavailability limits P utilization in plantations (including *Eucalyptus* plantations) in the acid soil area of southern China. Plants mainly utilize some low molecular organic P and inorganic P fractions, which are connected with microbial biomass turnover and originate from phosphatase hydrolysis, respectively [23–25]. A former report found microorganisms that live in low-TP conditions had a stronger ability to capture P from stable minerals [26]. These P fractions are immobilized in microbial biomass and absorbed by mycorrhizal-root associations during the growth-death cycle of microbes [27–29]. Meanwhile, microorganisms and plant fine root exude phosphatase enzymes for hydrolysis of stable organic P fractions; however, their bioactivity and efficiency cannot be maintained for a long time [23,24]. The synthesis and exudation of phosphate enzymes must be continued for microbes and plants to absorb inorganic P. These processes consume a lot of carbon (C), nitrogen (N), and other mineral nutrients [30] and are impacted by forest environmental changes, soil physico-chemical properties, and human activities [24,31], which suggests that some silviculture actions can influence soil P plant availability and available mechanisms [15,32,33]. Therefore, it is meaningful to study the effects of thinning on soil P bioavailability and cycling in high-density and short-rotation *Eucalyptus* plantations. Such research could potentially inhibit soil P deposition, increase the utilization efficiency of P fertilizers, and reduce pollution during production, shipping, and use of P fertilizers.

*Eucalyptus grandis* × *E. urophylla* is one of the most common species in short-rotation and high-density *Eucalyptus* plantations in the Guangxi Zhuang Autonomous Region of China. It is usually regenerated by the coppice method, which can lead to land degradation [34]. In this study, an *E. grandis* × *E. urophylla* plantation with an initial density of 1665 tree $ha^{-1}$ and 8 years of rotation was used as an experimental stand. We applied different thinning intensity treatments to a 6-year-old of the 2nd generation that was regenerated by coppice after clear cutting conducted on the 1st generation. Two years late (the age of the stands was 8 years old; it is the end of the second rotation), we collected soil samples from the 0–20 and 20–40 cm layers and measured soil nutrient content, microbial biomass, and enzyme activity. Specifically, two research questions were addressed: (1) Does thinning increase soil nutrient content and bioavailability, particularly P? (2) Which indicators play a significant role in soil P plant availability turnover in *E. grandis* × *E. urophylla* coppice forests? To answer these questions, three hypotheses were developed: **Hypothesis 1**: *Thinning improves soil AP content in E. grandis × E. urophylla coppice forests.* **Hypothesis 2**: *Microbial biomass phosphorus (MBP) is the most important factor affecting AP content in E. grandis × E. urophylla coppice forests.* **Hypothesis 3**: *Microbial biomass turnover is more important than enzyme hydrolysis for soil P bioavailability in*

*E. grandis* × *E. urophylla coppice forests*. This study aimed to provide insights into the effects of thinning on soil nutrient dynamics and to identify key indicators of soil P bioavailable turnover in *E. grandis* × *E. urophylla* coppice forests.

## 2. Materials and Methods

### 2.1. Study Site

The study was conducted at Qipo Forest Farm (108°43′–108°44′ E, 23°37′–23°38′ N) in Nanning city, Guangxi Zhuang Autonomous Region, within the southern subtropical hill area of China (Figure 1). The altitude of the study site is approximately 300 m. The climate in this region is subtropical monsoon, with annual mean, maximum, and minimum temperatures of 21.4 °C, 39.0 °C, and −2.2 °C, respectively. The annual total precipitation is 1300 mm and concentrates in the summer, with approximately 300 days of accumulated temperature over 10 °C per year and an annual total accumulated temperature of about 7200 °C. The dominant soil type in the area is Latosol, with a pH range of about 4.0–5.0, developed from granite or milestone, and a depth of about 80–90 cm. The understory plant community is composed of several common species, including *Miscanthus floridulus*, *Microlepia hancei*, *Pteris semipinnata*, *Rubus cochinchinensis,* and *Maesa japonica*.

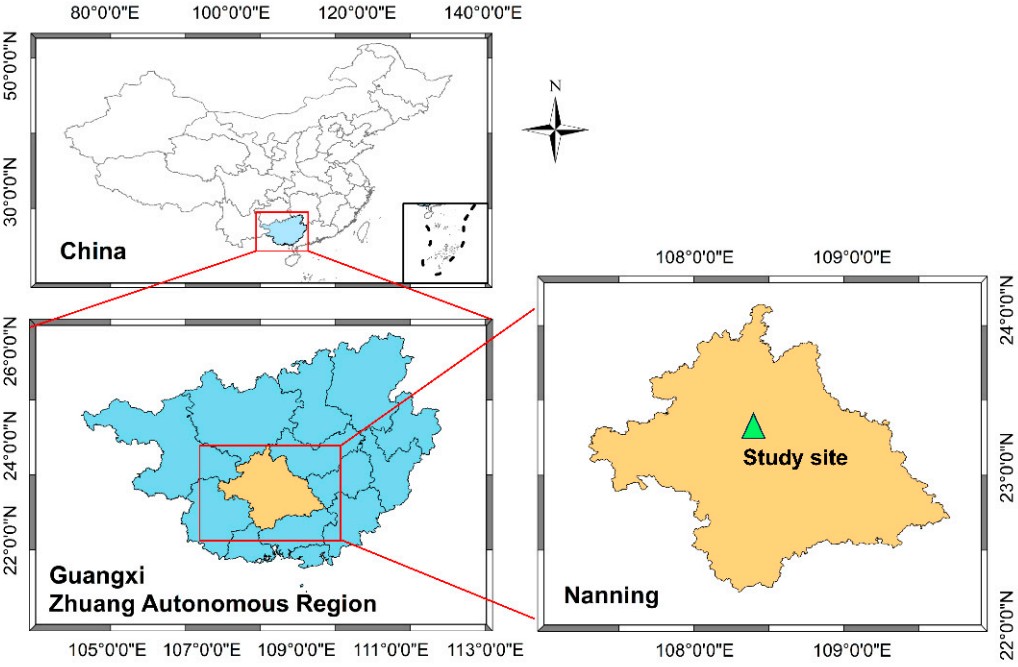

**Figure 1.** The location of the study site.

### 2.2. Experimental Design

The high density and short rotation *E. urophylla* × *E. grandis* DH32-28 experimental plantation located on a sloping land of sedimentary hill with an initial density of 1665 trees ha$^{-1}$ and 8 years as a rotation. This forest, 1st generation, was planted by saplings in 2005, and all of the timber was harvested by clear cutting in April 2013. Some silvicultural practices were used to regenerate: (1) stumps were protected by spraying disinfector to promote coppice budding when harvesting was complete; (2) after two months, the 3–4 stronger coppices were retained, which budded from different sides on a stump with similar height and diameter; meanwhile, other coppices were removed; and (3) two months later again, we selected the best one, which would be cultivated to 2nd stand, from the retained coppices and removed others on every stump. In every August during 2013–2015, 0.5 kg of NPK compound fertilizer was applied to every tree at four spots around the trunk; the distance from the fertilizing spots to the trunk was about 30 cm. In March 2019, we established fifteen 400 m$^2$ (20 × 20 m) plots with similar growth conditions, slop aspect,

slop gradient, and understory plant community at this 2nd *Eucalyptus* coppice (Table 1). The distance between each single plot was 30 m. Five thinning treatments with different intensities were assigned to fifteen plots through a completely randomized design. Each thinning-intensity treatment was replicated three times. The 5 thinning intensities were 83% (T1), 66% (T2), 50% (T3), 33% (T4), and 0% (CK), and the stand densities were 283, 566, 833, 1116, and 1665 tree ha$^{-1}$ after thinning, respectively (Table 1). We selectively thinned weakened trees under the canopy, and no logging slash, including root, bark, branch, and leaf, was removed. After thinning, no further silviculture activity was conducted, and a 10 m buffer zone was established around every plot.

**Table 1.** Basic information of *E. grandis* × *E. urophylla* coppices forest on different thinnings.

| Indexes | T1 | T2 | T3 | T4 | CK |
|---|---|---|---|---|---|
| Slop gradient (°) | 25 | 24 | 23 | 27 | 25 |
| Slop aspect | Northwest | Northwest | Northwest | Northwest | Northwest |
| Elevation (m) | 302.1 | 299.8 | 304.5 | 303.9 | 304.6 |
| Stand density (tree ha$^{-1}$) | 283 | 566 | 833 | 1116 | 1665 |
| Canopy density | 0.35 | 0.55 | 0.70 | 0.82 | 0.96 |
| Average tree height (m) | 26.23 | 23.87 | 23.58 | 25.37 | 24.67 |
| Average diameter at breast height (cm) | 24.17 | 19.53 | 19.60 | 19.43 | 16.77 |
| Stock volume (m$^3$ ha) | 113.86 | 148.41 | 199.56 | 259.27 | 365.60 |
| Soil bulk density (g cm$^{-3}$) | 1.28 | 1.31 | 1.38 | 1.41 | 1.47 |
| Soil total porosity (%) | 45.56 | 43.43 | 41.69 | 40.42 | 37.14 |
| Soil water content (%) | 14.14 | 15.18 | 12.08 | 11.96 | 12.60 |
| Soil pH | 4.29 | 4.09 | 4.17 | 4.11 | 4.25 |

Note: T1—83% intensity thinning; T2—66% intensity thinning; T3—50% intensity thinning; T4—33% intensity thinning; CK—Control.

### 2.3. Sampling and Analysis

In September 2021, we measured the growth characteristics of *Eucalyptus* trees and canopy densities at every plot (Table 1). Simultaneously, we set nine sample dots following an "S" pattern in the middle of planting lines within each plot according to the method outlined by Dang et al. [11]. We carefully removed the litter from the soil surface and subsequently collected one soil sample from each designated point in both the 0–20 and 20–40 cm layers. The nine soil cores within a plot and layer were mixed together to create a composite sample. A total of 30 mixed soil composites (5 treatments × 3 plots of a treatment × 2 layers) were stored in 4 °C boxes with dry ice. In the laboratory, we fully mixed all the soil composites again, removed any residual debris and litter, passed them through a 2 mm griddle, and then divided them into two sub-amples. One sub-sample was processed by air drying, grinding, and sieving using a 0.25 mm griddle for measuring abiotic indicators. The other sub-sample was stored at 4 °C in a refrigerator for biotic indicator determination.

SOC and TN were measured using an elemental analyzer (Vario EL III; Elementar, Frankfurt, Germany) [35]. Additionally, AN was determined using the previous method described by Wang et al. [36]. The soil samples were digested and processed by 0.5 mol NaOH, steam distillation in succession, and then titration with boric acid. To measure TP, the soil sample was digested with $HClO_4$-$H_2SO_4$ mixed liquor, and the supernatant was absorbed and analyzed using the molybdenum blue colorimetric method at 880 mm (UV-3600i Plus, SHIMADZU, Hadano, Japan) [37,38]. Furthermore, microbial biomass carbon, nitrogen, and phosphorus (MBC, MBN, and MBP) were analyzed using the chloroform fumigation-extraction method [37]. The soil samples were incubated in darkness in an incubator (MJ-250-I, Jiecheng Experimental Instrument CO, LTD, Shanghai, China) for 48 h and then fumigated and extracted before being extracted with 0.5 mol L$^{-1}$ $K_2SO_4$ for MBC and MBN and with Bray-1 (0.03 mol L$^{-1}$ $NH_4$F-0.025 mol L$^{-1}$ HCL) for MBP [39]. The oscillator shaking method (WSZ-100AR, Yiheng, Shanghai, China) was used during extraction. The C and N contents in the supernatant were measured using a TOC/TN

analyzer (Multi N/C 3100, Analytic Jena, Jena, Germany), and the P content was analyzed using the molybdenum blue colorimetric method (UV-3600i Plus, SHIMADZU, Japan). DOC and AP were measured as the C and P contents of the leaching solutions before fumigation, respectively [35]. A universal conversion factor of 0.45, 0.54, and 0.4 was used to calculate MBC, MBN, and MBP, respectively [40]. The quotients of microbial biomass carbon, nitrogen, and phosphorus (qMBC, qMBN, and qMBP) are expressed as the ratios of MBC:SOC, MBN:TN, and MBP:TP, respectively. The activities of soil β-glucosidase (BG) and acid phosphatase (ACP) were determined using the microplate technique as previously described by Saiya-Cork et al. [41]. Whereas the previous method described by Fei et al. was used to measure urease (URE) activity [42], it was presented by the ammoniacal N content of 5 g of fresh soil dissolved in 10 mL of urea solution (10%, *w/w*) and 10 mL of citrate buffer (pH = 6.7), incubated at 38 °C under dark conditions for 24 h.

### 2.4. Statistical Analysis

SPSS 18.0 (IBM, Chicago, IL, USA) was used for statistical analysis. A one-way of variance (ANOVA) was conducted to test statistical significance among treatments, followed by Duncan's multiple range test ($p < 0.05$) for various comparisons. The data were presented as mean ± standard difference. Meanwhile, the Pearson correlation was analyzed by this software. The ggplot2 in R version 4.2.1 and the R Studio 2022.07.0 interface were used for regression analysis. Based on the results of Pearson correlation, partial least squares path modeling (PLS-PM) was conducted using the pls-pm package to explore the direct or indirect connections of soil nutrients (including SOC, DOC, TN, and TP), enzymes of C and N cycling (including BG and URE), quotients of microbial biomass (including qMBC and qMBP), ACP, MBP, and AP [35,43]. Finally, we used Origin Pro 2022 (Originlab, Northampton, MA, USA) and Microsoft PowerPoint 2016 (Microsoft, Redmond, WA, USA) for drawing figures.

### 3. Results

*3.1. Response of Soil Nutrients Content to Thinning Intensity in E. urophylla × E. grandis Coppice Stands*

The soil nutrient content of different thinning treatments from different soil layers is presented in Table 2. Soil total nutrient content (SOC, TN, TP, and AN) increased and then decreased with increasing thinning intensity at the 0–20 cm layer. However, no significant differences ($p < 0.05$) were observed for these nutrients between the different thinned *E. urophylla × E. grandis* coppice stands in two layers. Soil TP in the 0–20 cm layer was significantly higher ($p < 0.05$) in the T1 treatment (81.41%) compared to the CK treatment, as well as in the T4 and T3 treatments (92.26%) (Table 2), but there were no significant differences ($p < 0.05$) observed in the 20–40 cm layer among the five treatments. This suggests that the stronger thinning intensities benefited the soil P content in the surface layer. In the 0–20 cm soil layer, the trend of AP content was observed as follows: T2 > T3 > T1 > CK > T4. Both T2 and T3 exhibited significantly higher ($p < 0.05$) AP levels than CK by 130.56% and 99.36%, respectively (Table 2). Similarly, these two treatments, T2 and T3, displayed significantly higher ($p < 0.05$) AP contents than T4 by 132.47% and 101.30%, respectively (Table 2). In the 20–40 cm layer, the trend was T2 > T1 > T3 > T4 > CK, with significantly ($p < 0.05$) increased soil AP content in the T2 treatment compared to the T4 and CK treatments by 142.31% and 215.00%, respectively (Table 2).

**Table 2.** Changes in soil nutrient content and the *p*-value and *F*-value of different thinned *E. urophylla* × *E. grandis* coppices forests.

| Indexes | Layers (cm) | T1 | T2 | T3 | T4 | CK | F | p |
|---|---|---|---|---|---|---|---|---|
| SOC | 0–20 | 11.62 ± 3.62 ns | 12.90 ± 2.51 ns | 10.72 ± 1.18 ns | 9.59 ± 1.30 ns | 9.46 ± 0.42 ns | 0.917 | 0.491 |
| (g kg$^{-1}$) | 20–40 | 8.55 ± 2.58 ns | 7.06 ± 1.66 ns | 6.56 ± 0.13 ns | 6.16 ± 0.10 ns | 7.37 ± 1.40 ns | 0.866 | 0.516 |
| DOC | 0–20 | 20.40 ± 5.48 ns | 23.53 ± 5.17 ns | 18.01 ± 5.05 ns | 12.79 ± 3.37 ns | 17.74 ± 1.38 ns | 1.733 | 0.219 |
| (mg kg$^{-1}$) | 20–40 | 8.89 ± 2.43 ns | 7.75 ± 1.00 ns | 10.11 ± 3.16 ns | 5.54 ± 0.83 ns | 7.87 ± 2.05 ns | 1.307 | 0.332 |
| TN | 0–20 | 0.74 ± 0.19 ns | 0.78 ± 0.15 ns | 0.63 ± 0.04 ns | 0.60 ± 0.09 ns | 0.61 ± 0.04 ns | 0.987 | 0.457 |
| (g kg$^{-1}$) | 20–40 | 0.62 ± 012 ns | 0.49 ± 0.13 ns | 0.47 ± 0.01 ns | 0.48 ± 0.02 ns | 0.55 ± 0.05 ns | 1.170 | 0.381 |
| AN | 0–20 | 38.76 ± 5.35 ns | 37.73 ± 7.41 ns | 34.65 ± 1.09 ns | 33.88 ± 1.09 ns | 34.39 ± 0.36 ns | 0.564 | 0.694 |
| (mg kg$^{-1}$) | 20–40 | 34.14 ± 7.23 ns | 30.29 ± 6.42 ns | 26.44 ± 0.96 ns | 24.90 ± 1.45 ns | 33.11 ± 1.66 ns | 1.645 | 0.238 |
| TP | 0–20 | 0.38 ± 0.08 a | 0.31 ± 0.09 ab | 0.20 ± 0.02 b | 0.20 ± 0.02 b | 0.21 ± 0.05 b | 3.573 | 0.047 * |
| (g kg$^{-1}$) | 20–40 | 0.26 ± 0.07 ns | 0.21 ± 0.06 ns | 0.18 ± 0.03 ns | 0.19 ± 0.00 ns | 0.20 ± 0.02 ns | 0.821 | 0.541 |
| AP | 0–20 | 1.29 ± 0.40 bc | 1.79 ± 0.38 a | 1.55 ± 0.27 ab | 0.77 ± 0.07 c | 0.78 ± 0.19 c | 5.044 | 0.017 * |
| (mg kg$^{-1}$) | 20–40 | 0.93 ± 0.27 ab | 1.26 ± 0.42 a | 0.76 ± 0.27 ab | 0.52 ± 0.07 b | 0.40 ± 0.09 b | 3.498 | 0.049 * |

Note: The lowercase letter indicates a significant difference (*p* < 0.05) between different thinning treatments in the same layer. T1—83% intensity thinning; T2—66% intensity thinning; T3—50 intensity thinning; T4—33% intensity thinning; CK—Control; SOC—Soil organic carbon; DOC—Dissolved organic carbon; TN—Total nitrogen; AN—Alkaline hydrolysable nitrogen; TP—Total phosphorus; AP—Available phosphorus; * and ns—significant difference at *p* < 0.05 and non-significant, respectively. (mean ± SD, *n* = 3).

*3.2. Response of Soil Microbial Biomass to Thinning Intensity in E. urophylla × E. grandis Coppice Stands*

The qMBC and MBC in soil showed the same trend (decreased and then increased with an increase in thinning intensity) among treatments except for the 20–40 cm layer of T4 (Figure 2a,b). Among soil layers, MBC was enhanced in 0–20 cm soil layers across thinning treatments (Figure 2a). Whereas qMBC showed the highest value in the 20–40 cm layer among all thinning treatments (Figure 2b). Among the treatments, T2 treatment had the lowest qMBC among all thinning treatments, with a significant difference (*p* < 0.05) of 47.34% and 44.81% compared to CK in the 0–20 cm and 20–40 cm layers, respectively (Figure 2b, Table 3). Moreover, T2 resulted in the lowest qMBN, which was significantly lower (*p* < 0.05) than T1, T3, T4, and CK by 39.13%, 36.04%, 40.09%, and 36.51%, respectively (Figure 2d, Table 3). Interestingly, soil MBP and qMBP in the 20–40 cm layer showed an increase followed by a decrease with stand density increasing. T2 was significantly higher (*p* < 0.01) than other treatments, showing an increase of 91.23%–189.45% in MBP and 67.55%–211.47% in qMBP (Figure 2e,f, Table 3). This suggests that T2 benefits soil P accumulation in microorganisms. However, thinning had no significant influence (*p* < 0.05) on MBN, MBP, and qMBP in the 0–20 cm layer (Figure 2c,e,f, Table 3), or on qMBN in the 20–40 cm layer (Figure 2d, Table 3).

**Table 3.** The *F*-value and *p*-value of one-way ANOVA of MBC, qMBC, MBN, qMBN, MBP, qMBP, BG, URE, and ACP in different thinning treatments and layers.

| Indexes | Layers (cm) | F | p |
|---|---|---|---|
| MBC | 0–20 | 8.528 | 0.003 ** |
| | 20–40 | 15.242 | <0.001 ** |
| qMBC | 0–20 | 5.202 | 0.016 * |
| | 20–40 | 5.782 | 0.011 * |
| MBN | 0–20 | 1.239 | 0.355 |
| | 20–40 | 0.403 | 0.802 |
| qMBN | 0–20 | 4.163 | 0.031 * |
| | 20–40 | 0.613 | 0.663 |

**Table 3.** *Cont.*

| Indexes | Layers (cm) | *F* | *p* |
|---|---|---|---|
| MBP | 0–20 | 2.155 | 0.148 |
| | 20–40 | 8.875 | 0.003 ** |
| qMBP | 0–20 | 0.191 | 0.937 |
| | 20–40 | 8.046 | 0.004 ** |
| BG | 0–20 | 3.604 | 0.046 * |
| | 20–40 | 3.892 | 0.037 * |
| URE | 0–20 | 0.344 | 0.409 |
| | 20–40 | 1.416 | 0.298 |
| ACP | 0–20 | 0.989 | 0.457 |
| | 20–40 | 1.232 | 0.357 |

Note: * and ** stand significance differences at *p* < 0.05 and *p* < 0.01, respectively.

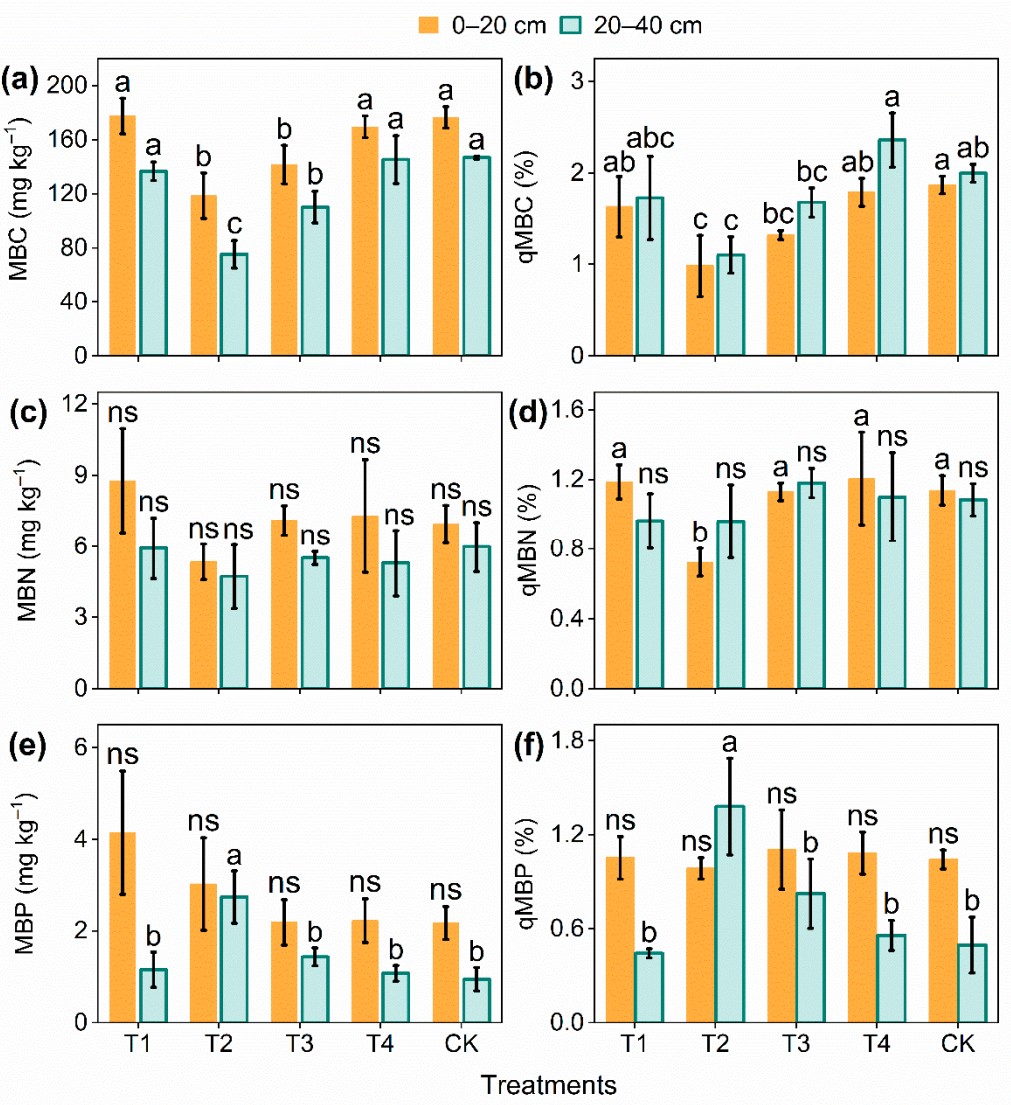

**Figure 2.** Changes in soil microbial biomass carbon (MBC) (**a**), nitrogen (MBN) (**c**), phosphorus (MBP) (**e**), the quotient of microbial biomass carbon (qMBC) (**b**), nitrogen (qMBN) (**d**), and phosphorus (qMBP) (**f**) of different thinned *E. grandis* × *E. urophylla* coppices forests (mean ± SD, *n* = 3). Note: ns—no significant difference. Different lower-case letters indicate significant differences among the treatments (*p* < 0.05).

### 3.3. Response of Soil Enzyme Activities to Thinning Intensity in E. urophylla × E. grandis Coppice Stands

Among soil enzyme activities, only BG activity was significantly affected ($p < 0.05$) by various thinning treatments (Figure 3a, Table 3), while the URE and ACP activities were not significantly ($p < 0.05$) influenced (Figure 3b, c, Table 3). BG activity was increased first and then decreased as thinning intensity increased, following the order T2 > T3 > T1 > CK > T4 and T2 > T3 > T4 > CK > T1 in 0–20 cm and 20–40 cm layers, respectively (Figure 3a). BG activity in T2 was higher than other treatments by 16.37%–40.35% in the 0–20 cm layer and by 10.10%–63.22% in the 20–40 cm layer (Figure 3a).

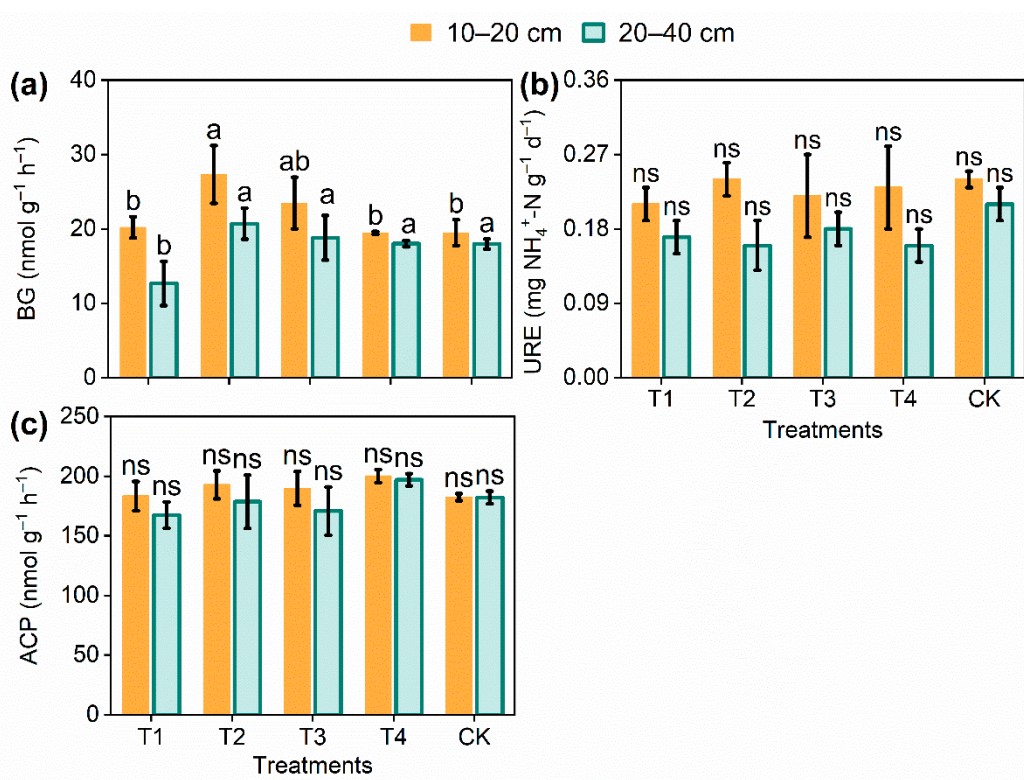

**Figure 3.** Effects of various thinning intensities on soil β-Glucosidase (BG) (**a**), urease (URE) (**b**), and acid phosphatase (ACP) (**c**) activities in *E. grandis* × *E. urophylla* coppices forests (mean ± SD, $n = 3$). Note: ns: no significant difference. Different lower-case letters indicate significant differences among the treatments ($p < 0.05$).

### 3.4. The Influence of Soil Nutrients, Microbial Biomass, and Enzyme Activity on P Availability

Regression and Pearson correlation analysis indicated that AP was significantly positively correlated with SOC ($R^2 = 0.476$), DOC ($R^2 = 0.457$), TN ($R^2 = 0.413$), AN ($R^2 = 0.404$), TP ($R^2 = 0.353$), BG ($R^2 = 0.425$), MBP ($R^2 = 0.48$) and qMBP ($R^2 = 0.203$) ($p < 0.05$) (Figures 4a–f and 5c,f, Table S1), while significantly negatively correlated with qMBC ($R^2 = 0.636$) and qMBN ($R^2 = 0.143$) ($p < 0.05$) (Figure 5d,e, Table S1). These results suggested that soil AP was affected by soil nutrients and microbial biomass indicators in *E. grandis* × *E. urophylla* coppice forests. Increasing nutrient supply and qMBP can promote soil P turnover from a stable to a bioavailable fraction.

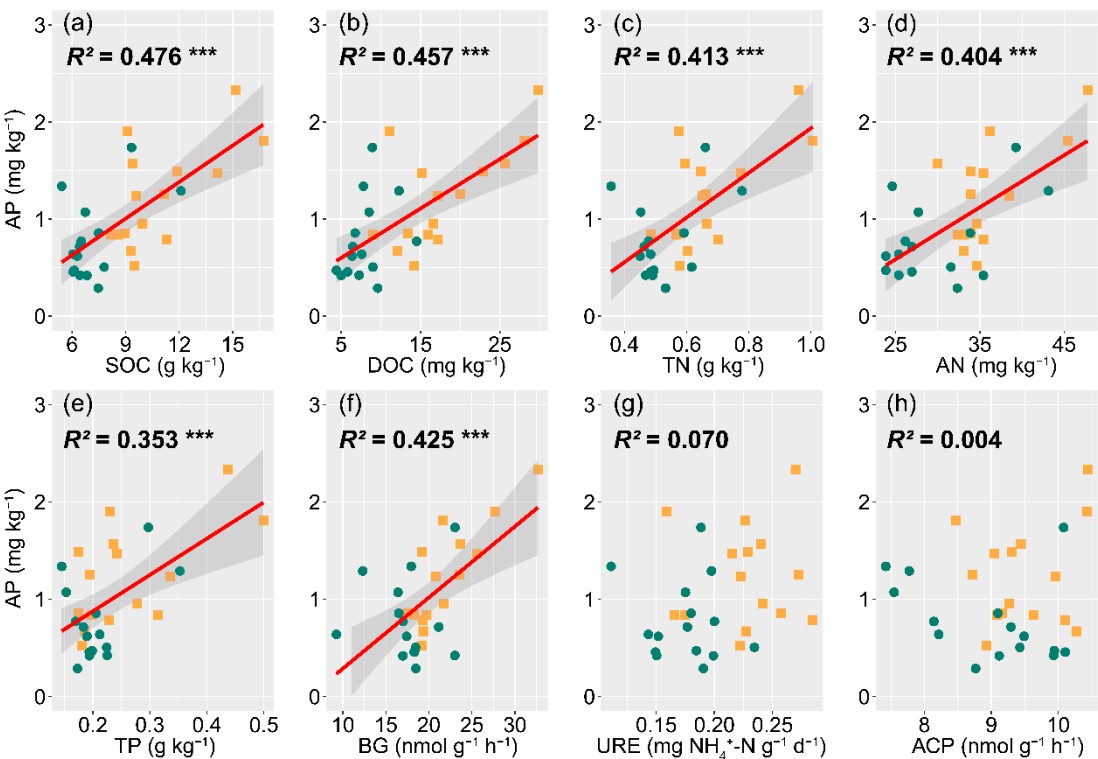

**Figure 4.** Regression analysis between AP and soil nutrients (**a**–**e**) and enzyme activities (**f**–**h**). Note: ***: $p < 0.001$. The confidence interval and fitting line were revealed at $p < 0.05$. Orange and depth green points were data in the 0–20 and 20–40 cm layers, respectively.

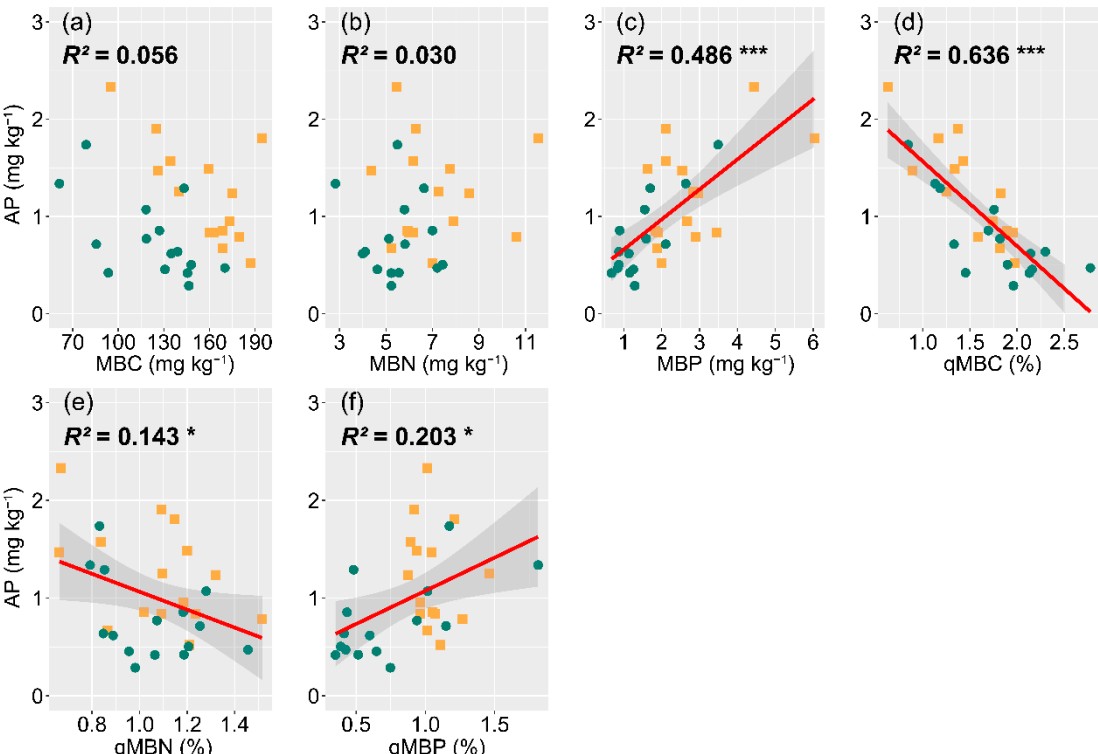

**Figure 5.** Regression analysis between AP and to microbial biomass (**a**–**c**) and quotient of microbial biomass (**d**–**f**). Note: *: $p < 0.05$, ***: $p < 0.001$. The confidence interval and fitting line were revealed at $p < 0.05$. Orange and depth green points were data in 0–20 and 20–40 cm layers, respectively.

The PLS-PM model revealed that soil nutrients had the highest standardized effect on AP, as shown in Figure 6b. The direct and indirect effects of soil nutrients through MBP, ACP, and the quotient of microbial biomass on AP were significant ($p < 0.05$) (Figure 6a). These results imply soil nutrient content was the most important factor affecting P bioavailability in *Eucalyptus* coppice forest. The quotient of microbial biomass was significantly affected by soil nutrients ($p < 0.05$), an enzyme of C and N cycling ($p < 0.01$), ACP ($p < 0.01$), and MBP ($p < 0.001$) directly (Figure 6a). However, the direct effects of enzymes C and N cycling, ACP, and MBP on AP were not significant ($p < 0.05$) (Figure 6a). Meanwhile, AP was significantly ($p < 0.01$) directly affected by the quotient of microbial biomass, suggesting that the effects of enzymes of C and N cycling, ACP, and MBP on AP were mainly through an indirect pathway (Figure 6). The two parameters of the quotient of microbial biomass, qMBC and qMBP, had different influences on AP, with qMBC having an adverse effect and qMBP having a positive impact (Figures 5d,f and 6a, Table S1). These results indicate that different indicators of microbial quotient had a varied influence on soil AP in an *E. grandis × E. urophylla* coppice forest.

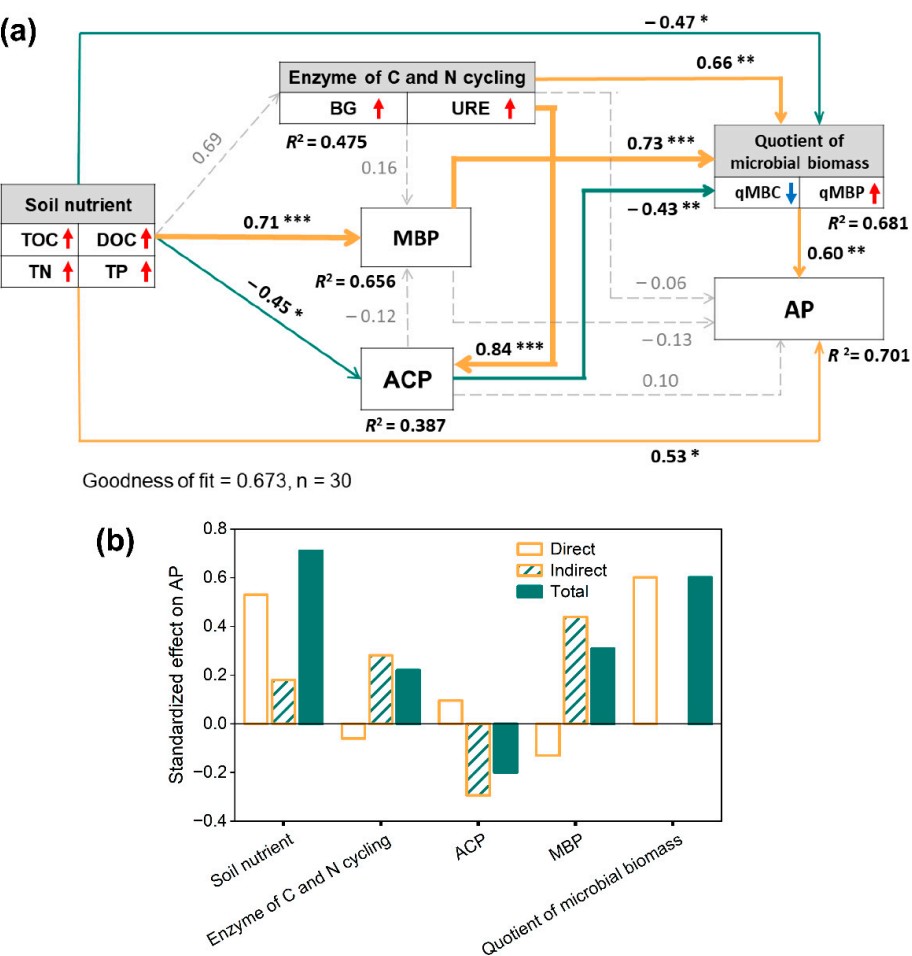

**Figure 6.** Partial least squares path model (PLS-PM) of soil nutrient, enzyme activity, microbial biomass, quotient of microbial biomass, and AP (**a**), and standardized direct, indirect, and total effect on AP of these indicators (**b**). Note: The numbers of the arrowed lines are path coefficients. Orange and deep green solid arrows indicate positive and negative significant ($p < 0.05$) paths, respectively. Gray dotted arrows represent a non-significant ($p < 0.05$) path. Red and blue arrows indicate positive and negative relationships, respectively, between measured variables and associated variables in measurement models of PLS-PM. *: $p < 0.05$, **: $p < 0.01$, ***: $p < 0.001$.

## 4. Discussion

### 4.1. Soil Nutrients of Different Intensity Thinned E. grandis × E. urophylla Coppice Plantations

Previous studies suggested that increasing soil SOC and TN after thinning requires an extended period [44,45]. In the present study, soil SOC, DOC, TN, and AN were not significantly influenced ($p < 0.05$) (Table 2) in different thinned *E. grandis* × *E. urophylla* coppice forests after two years. These results confirmed the regular fitting of the *Eucalyptus* plantation in Guangxi. Thinning of *Eucalyptus* plantations in short rotation can lead to a decrease in stand density, an increase in canopy openness, and destruction of the understory plant community and soil structure, which can accelerate the loss of soil C and nutrients, especially some labile C and N fractions, due to an increase in soil temperature, priming effect, and enzyme activity [27,46–48]. In the long term, thinning benefits plant community structure and promotes the soil nutrient cycle [11]. However, thinning provides *Eucalyptus* trees with more space for growth, enhancing their photosynthesis and nutrient uptake. Meanwhile, 6-year-old to 8-year-old *Eucalyptus* trees in the current study were in a rapid growth period; these accelerated soil nutrients utilized by trees and reduced litter input [49], which prolonged the process of increasing soil C and N contents in thinned *Eucalyptus* plantations. The SOC, DOC, TN, and AN trends were varied in two layers. These four indicators were higher in the T2 and T3 treatments than CK in the 0–20 cm layer but lower in the 20–40 cm layer (Table 2). The results comply with the rule of soil nutrients content in different layers that aggregate on the soil surface in subtropical plantations [50,51] and may be related to spatial variation of litter input and organism nutrients demand: (1) the understory vegetation root system may not develop entirely in these stands, and litter is mainly concentrated on the soil surface [52]; (2) meanwhile, *Eucalyptus* trees usually uptake nutrients from the depth of the soil layer, and microorganisms indicate higher N demand in soil below 20 cm in subtropical forests [50]. Therefore, the amount of soil nutrients consumed is higher in the 20–40 cm layer compared to the 0–20 cm layer.

Qiao et al. reported that the average TP content of native broad-leaved forest in the 0–20 cm and 20–40 cm layers was $0.38 \pm 0.12$ g kg$^{-1}$ and $0.31 \pm 0.11$ g kg$^{-1}$, respectively, in Tiantong National Park, a subtropical region similar to our study area [53]. Our findings indicated that the soil TP contents of the CK treatment were $0.21 \pm 0.05$ g kg$^{-1}$ and $0.20 \pm 0.02$ g kg$^{-1}$ in these two layers (Table 2). This suggests that soil P accumulations in our 2nd generation *E. grandis* × *E. urophylla* coppice plantation were lower than in the native forest ecosystem. It is well known that there is more soil nutrient consumption and erosion in short-rotation and high-density *Eucalyptus* plantations than in native forest ecosystems [4,54,55]. Soil TP increased in both the T1 and T2 treatments after two years (Table 2), suggesting that thinning, especially high-intensity thinning, can promote soil P levels. Bedrock and mineral conditions are essential factors that affect soil TP content [56], indicating that TP is relatively stable in a specific ecosystem and that limited research exists. Therefore, litter input is an essential source for the soil P pool [20]. Our study found that soil TP is significantly positively ($p < 0.001$) related to SOC (Table S1), which suggests that increased litter input is the main reason for soil P accumulation in the 0–20 cm layer of T1 and T2 treatments. Soil AP content in the T2 and 0–20 cm layer of T3 was significantly higher than in the CK ($p < 0.05$) (Table 2), suggesting that moderate thinning intensities can improve P plant availability in *E. grandis* × *E. urophylla* coppice forest, thus confirming that thinning can improve soil AP content in *E. grandis* × *E. urophylla* coppice forest.

### 4.2. Soil Microbial Biomass and Enzyme Activity of Different Intensity Thinned E. grandis × E. urophylla Coppice Plantations

The results showed that soil MBN and MBP were significantly positively correlated ($p < 0.01$) with soil SOC, DOC, TN, AN, and TP (Table S1). However, MBC did not show a significant correlation ($p < 0.05$) with these soil nutrients (Table S1), suggesting that MBN and MBP were controlled by soil nutrient supply, while MBC was impacted by changes in the understory plant community and root traits after thinning [27,57]. The changes in plant structure after thinning were found to be a major reason affecting soil



fungi [58,59], such as community, distribution, structure, and physiological activity. Fungi have a higher C proportion than bacteria in the soil and are the main fraction of soil MBC [11,60,61]; hence, the MBC of our stand soil changed after thinning. On a global scale, the average qMBN and qMBP in tropical and subtropical forest soils are 3.08% and 6.32%, respectively [62]. In our study, the soil average qMBN and qMBP of *E. grandis* × *E. urophylla* coppice plantations were 1.07% and 0.90%, respectively (Figure 2d,f), which are only a part of 34.74% and 14.24%, respectively, on the global scale. These results may be related to worse soil nutrient loss, physico-chemical properties, microbial multifunctionality, and increasing generations in *Eucalyptus* plantations [5,63,64]. This implied that the proportions of MBN and MBP in soil TN and TP were lower. Regression, Pearson correlation, and PLS-PM analysis suggested a significant positive relationship between qMBP and AP ($p < 0.05$) (Figures 5f and 6, Table S1). These results implied that the relatively inferior qMBP was a reason for low P bioavailability [23].

Soil BG activities rose initially and then decreased with thinning intensity in the current study (Figure 3a); this trend was opposite to the findings of Chen et al., who reported lower BG activity in low-intensity thinning treatment and higher activity in high-intensity treatment than CK in Chinese fir plantations [65]. These varied trends in BG activities suggest that tree species are one of the reasons affecting enzyme activity in thinned plantations. There are different forest structures, light environments, plant communities, and soil nutrient cycling characteristics in these plantations, which are constructed by various broad-leaved or coniferous tree species [66,67]. We found that BG activity had a very significant positive correlation ($p < 0.01$) with SOC and DOC (Table S1), suggesting that it was affected by substrate quantity and quality [68]. However, according to the results found by Zhou et al., N and P hydrolytic enzyme activity increased after thinning in the larch (*Larix olgensis*) plantation [48]. While our study found no significant difference ($p < 0.05$) in URE and ACP activity in *E. grandis* × *E. urophylla* coppice forest (Figure 3b,c), this result was reported in other tropical and subtropical plantations, such as teak (*Tectona grandis*) [69]. In nutrient-faulty acid soils, N and P absorbed by Fe and Al ions limit hydrolytic reactions of URE and ACP, and uptake of N and P from soil organic matter through mycorrhiza is a more efficient way for plants [19,22]. Therefore, URE and ACP activity in bulk soil may not be essential indicators for N and P uptake in *E. grandis* × *E. urophylla*, and matched studies between rhizosphere and bulk soils should be conducted in *Eucalyptus* plantations.

*4.3. Function of Soil Total Nutrient and DOC on P Bioavailability in E. grandis × E. urophylla Coppice Forest*

We observed that the standardized total effect on AP of soil nutrients (SOC, DOC, TN, and TP) was higher than enzyme activities, MBP, and the quotient of microbial biomass (Figure 6b). These results suggested soil total nutrients and DOC contents were the most important indicators for P bioavailability. Furthermore, C, N, and P are major elements required for microbial activity, especially heterotrophs. Adequate soil nutrient supply promotes continual turnover of soil nutrients and enzyme activity, thereby accelerating soil P plant availability [14,31]. Additionally, higher soil total nutrient levels promote plant growth, leading to improved litter input and long-term maintenance of forest soil fertility [70–72]. Moreover, DOC is the most effective energy resource for soil microbes and can affect soil microbial community and function [73]. Previously, several studies reported different effects of soil total nutrient and DOC on the soil P biochemical cycle; for example, Margalef et al. (2017) documented that TN was the principal factor that influenced the spatial gradient change of phosphatase activity on a global scale [31], and Zhang et al. found that long-term fertilizer input improved soil SOC and TN contents, resulting in increased MBP in the Loess Plateau [74]. Some fractions of DOC, such as low-molecular-weight organic acids, can occupy or change positions of soil particles that bind inorganic P, inhibiting the absorption of bioavailable P fractions and consequently increasing soil AP content [75].

*4.4. Effects of Soil Microbial Biomass Turnover and Enzyme Activity on P Availability*

Extracellular enzyme activities and microbial biomass turnover are two major pathways of soil P bioavailability [19,31]. Through regression, Pearson correlation, and PLS-PM analysis, we found that the relationship between MBP and AP was stronger than that between ACP and AP (Figures 4h, 5c and 6a, Table S1). Furthermore, the total standardized effects on AP of MBP and ACP were 0.31 and 0.20, respectively (Figure 6b), suggesting that MBP turnover had a stronger influence on AP than ACP in 2nd generation *Eucalyptus* coppice plantations [76]. Soil AP is utilized by microbes for their metabolism [77], but we found that the direct standardized effect on AP of MBP was not significant ($p < 0.05$) in our *Eucalyptus* plantation (Figure 6a). These results confirmed that microbial biomass turnover was more important than enzyme hydrolysis for soil AP in *E. grandis* × *E. urophylla* coppice forest. Additionally, the direct standardized effect of MBP on AP was negative and not significant ($p < 0.05$), but its indirect standardized effect on AP was positive, and the pathway MBP—the quotient of microbial biomass—AP was highly significant ($p < 0.01$) (Figure 6). This implies that regulating qMBP was the main way that MBP affected AP in the *Eucalyptus* coppice plantation. These results highlighted that accelerating P relative accumulation in microbial biomass effectively promotes soil P bioavailable turnover.

The effects of two quotients of microbial biomass indicators on AP were varied, with qMBC and qMBP having a significant negative correlation ($p < 0.01$) and a significant positive correlation ($p < 0.05$) with AP, respectively (Figures 5d,f and 6a, Table S1). These results suggest that the increases in MBC to SOC ratio and MBP to TP ratio had negative and positive effects on soil P bioavailability, respectively, in our *Eucalyptus* stands. On the one hand, C is a restricted element for soil microbes, according to former studies. Soil microbe activity and priming effect are maintained by an abundant C supply. Conversely, soil microbes are in inactive dormancy if soil C is insufficient, so soil nutrients cycle and turnover through microbial biomass are controlled by SOC contents [14,78]. The qMBC is an important indicator that reflects the ability of soil C supply; a lower qMBC means specific quantities of microbes can gain more C sources relatively from soil [79,80], and then some activities related to soil P bioavailability, which are mediated by microbial turnover, can be sustained [24,81]. Therefore, the relationship between qMBC and AP was significantly negative ($p < 0.001$) (Figure 5d, Table S1). We found that T2 treatment and the 0–20 cm layer of T3 treatment were significantly lower ($p < 0.05$) than T4 and CK treatments in qMBC (Figure 2b). These results imply that soil microbes can gain more energy resources from these treatments and layers, which benefits soil P bioavailability. Soil AP content was significantly enhanced ($p < 0.05$) in these two treatments (Table 2). On the other hand, soil microbes can uptake organic P and immobilize it in microbial biomass as nucleic acid, phosphatide, ATP, etc., and return it to the soil in small molecular organic P such as metabolic exude and residual body. These can be taken up by plant-mycorrhizal fungi systems [24,82,83]. Furthermore, this suggests that enhancing the MBP-to-TP ratio can promote soil P bioavailable turnover. This is the major reason for the significant positive relationship of qMBP with AP in *Eucalyptus* coppice plantations (Figure 5f, Table S1).

The regulation of the global soil P cycle and bioavailability are regulated by hydrolytic enzymes; however, the function of hydrolytic enzymes varies depending on the ecosystem [68,77,84]. For example, Zhu et al. found that standardized total effects on bioavailable-P of ACP were negative in subalpine coniferous (*Abies fabri*) forests in the eastern edge of the Tibetan Plateau [68]. This phenomenon also emerged in our study and the *E. grandis* × *E. urophylla* coppice forest (Figure 6). Lu et al. conducted a county-scale study in China to determine the role of biotic factors on soil bioavailable P fraction, and they concluded that other hydrolytic enzymes, in addition to ACP, that link with soil P mineral processes, e.g., phytase, were in an essential location in soil P bioavailable turnover, especially in some land use types that were disturbed by human activities [56]. Short rotation and frequent fertilizer input make *E. grandis* × *E. urophylla* plantations an ecosystem seriously influenced by industrialized forest management. Therefore, using ACP alone may not accurately reflect the function of hydrolytic enzymes related to soil

P mineralization. According to Lu et al., this was the main reason soil ACP showed no significant difference ($p < 0.05$) along five intensities of thinning treatments (Figure 3c) [56]. Furthermore, we found soil BG activity had a significant ($p < 0.01$) positive relationship with AP (Figure 4f, Table S1). In the PLS-PM model, the enzymes involved in C and N cycles (BG and URE) were key variations on AP through ACP and the quotient of microbial biomass (Figure 6). This suggests that increasing BG and URE activities can indirectly promote soil P bioavailability in *Eucalyptus* plantations. High BG and URE activities can (1) supply sufficient nutrients for soil microbes [14], (2) promote soil P accumulation in microbial biomass and enzyme synthesis [85], and (3) increase the microbial turnover rate [76]. These results suggested a tight relationship between C, N, and P cycles in the soil of *E. grandis* × *E. urophylla* coppice plantations, which has been confirmed in grassland and cropland [27,85–87].

## 5. Conclusions

In conclusion, our study found no significant differences ($p < 0.05$) in SOC, DOC, TN, and AN among five thinned-intensity treatments. However, we observed that the T1 stand had a significantly higher ($p < 0.05$) TP content in the 0–20 cm layer; meanwhile, the T2 and 20–40 cm layers of T3 had significantly higher ($p < 0.05$) AP contents. Additionally, our analysis implies that various soil indicators can be used to assess P bioavailability. Firstly, soil SOC, TN, TP, and DOC contents had the highest standardized effect on AP. Secondly, microbial turnover was the key process in the soil P cycle, and increasing qMBP benefited P bioavailability, while lower qMBP restricted soil AP content in *Eucalyptus* plantations. Lastly, soil P plant bioavailability is closely connected with soil C and N supply and cycle. These findings, which confirmed hypotheses 1 and 3, and rejected hypotheses 2, suggest that relatively heavy thinned intensity can accelerate soil P content and bioavailable turnover and highlight the primary functions of microbial biomass turnover and soil nutrient supply in the P plant-available mechanism in *Eucalyptus* coppice plantations. Therefore, some silviculture practices, such as prolonging rotation and introducing broad leaf or nitrogen-fixing tree species, should be used to inhibit soil nutrients loss and increase litter input for maintaining soil nutrient bioavailable turnover after thinning. More future research should focus on the influences of thinning soil nutrients and biochemical processes in the long term.

**Supplementary Materials:** The following supporting information can be downloaded at: https://www.mdpi.com/article/10.3390/f14102067/s1, Table S1. Pearson correlation matrix of soil nutrient, microbial biomass, and enzyme activity in *E. grandis* × *E. urophylla* sprout forests.

**Author Contributions:** Funding acquisition, S.Y. and M.Y.; Investigation, X.D. and Y.X.; Writing—original draft, X.X.; Writing—review and editing, I.A. All authors have read and agreed to the published version of the manuscript.

**Funding:** This work was supported by the National Natural Science Foundation of China (32371856) and the Key Projects of Guangxi Natural Science Foundation (2021GXNSFDA196003).

**Data Availability Statement:** Data will be made available upon request.

**Acknowledgments:** This work was supported by National Natural Science Foundation of China (32371856) and Key Projects of Guangxi Natural Science Foundation (2021GXNSFDA196003). We sincerely thank Qipo Forest Farm which gave help in plots selecting and soil sampling. And we sincerely thank Hongxiang Wang for beneficial advice in writing.

**Conflicts of Interest:** The authors have no relevant financial or non-financial interests to disclose.

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
