# Peer review of "Thinning Promotes Soil Phosphorus Bioavailability in Short-Rotation and High-Density Eucalyptus grandis × E. urophylla Coppice Plantation in Guangxi, Southern China"

_forests, doi:10.3390/f14102067_

Round 1

Reviewer 1 Report

The manuscript corresponds to the theme of the Special Issue. The manuscript presents the results of a study revealing the mechanisms of changes in the availability of soil phosphorus in plantation soils depending on plant density. The results are adequately processed, well presented, and support the conclusions. In the Discussion, the authors attempted to suggest plausible mechanisms underlying the observed phenomena.

Below are a few specific points that need to be improved before publication.

In my opinion, the Introduction could be shortened somewhat. It partially contains generally known information that does not lead the reader to the research hypotheses.

L115: Please use the term "precipitation" instead of "moisture"

L136: Missed verb. Should be "the distance ... WAS... "

L177: 0.45 is a conversion factor for MBC only.  For MBN standard conversion factor is 0.54 [Brookes et al., 1985]

L178: Did you use a bystander (soil sample with the addition of a known amount of phosphate before extraction) to determine the conversion factor for microbial biomass phosphorus? This factor can vary significantly depending on the content of iron, aluminum, and other chemical compounds in the soil that actively sorb phosphates.

Fig. 1 - Typo. Should be "TrEAtments"

L491: Please use the term "content" instead of "concentration" to refer to soil compounds. Concentration is a term applicable only to liquids and gases; for solids (such as soil) it is more correct to use the term "content"

In my opinion, a great idea would be to discuss in the Conclusion which of the hypotheses were confirmed and which were not. You formulated good hypotheses at the beginning of the manuscript but completely forgot about them.

Reviewer 2 Report

In tropical forests, phosphorus cycling is known to be concentrated in the epiphytic community rather than in red-colored weathered soils. On the other hand, secondary afforestation of tropical areas is of particular importance and eucalypts are used for this purpose. This is not an epiphytic ecosystem, but a regular forest ecosystem. In this case there is a reorganization of biogeochemical processes back to red-colored terrestrial soils. Therefore, the study of the formation of phosphorus status of soils under soil plantations is highly demanded.

In connection with the above, I recommend citing also works on phosphorus cycling in mature tropical forests with a pronounced epiphytic understory. Another remark concerns the interpretation of data - the article contains a lot of statistical evidence and correctly calculated coefficients, including correlation coefficients. At the same time, very little attention is paid to the actual soil-genetic interpretation of soil processes.

Minor comments:

What is “breakdown of soil structure” – may be degradation of aggregates?

I don’t agree with this statement “In low latitude regions, soil bioavailable P is relatively scarce due to weathered bed rock and inorganic P adsorbed by Fe and Al ions” – weathered not equal to leached!, clarify, please.

Line 111 – it is necessary to clarify natural zone, type of landform and type of hills.

I recommend to provide the photo of Latosol, photo of vegetation cover, as well as insert map of the study sites.

I understand that soil were investigated in terms of chemistry in the fine earth fraction, but is it possible to provide data on ration of skeletal fraction and fine earth?

I recommend to discuss texture class of soil as well as some data on mineralogical composition cause it is important in terms of phosphorous regime.

Reviewer 3 Report

Dear Editor,

I would like to thank you for your confidence in reviewing this manuscript.

I send you here my comments for the manuscript review.

Title: Thinning promotes soil phosphorus bioavailability in short-rotation and high-density Eucalyptus grandis × E. urophylla coppice plantation in Guangxi, Southern China.

Remarks/suggestions

The manuscript is well written and presented regarding the different sections. I recommend its publication after minor revisions on English Language.

References:

Please check references (in text and list) in relation to the journal's recommendations.

Author Response

Thank your positive comments. We have carefully checked the manuscript, include references, and corrected it according to the comments of you and others two reviewers.